# Formulation of a Gastroretentive In Situ Oral Gel Containing Metformin HCl Based on DoE

**DOI:** 10.3390/pharmaceutics14091777

**Published:** 2022-08-25

**Authors:** Jong Hee Kim, Seung Hyun Song, Sang Hoon Joo, Gyu Hwan Park, Kwon-Yeon Weon

**Affiliations:** 1College of Pharmacy, Daegu Catholic University, Gyeongsan 38430, Korea; 2College of Pharmacy, Research Institute of Pharmaceutical Sciences, Kyungpook National University, Daegu 41566, Korea

**Keywords:** metformin HCl, in situ oral gel, experimental design, formulation, sustained release (SR)

## Abstract

A gastroretentive in situ oral gel containing metformin hydrochloride (Met HCl) was prepared based on sodium alginate (Sod ALG), calcium carbonate, and hydroxyethylcellulose (HEC). The optimal composition of the formulation was explored based on the design of experiments (DoE). First, a 3^2^ full factorial design was used for formulation E1 to determine proper composition of Sod ALG and calcium carbonate. Second, a circumscribed central composite design was employed to add HEC as a thickening agent (formulation E2). The dissolution rates at 15, 30, 60, 120, and 240 min were used as responses. Partial least squares regression analysis indicated the effect of each component in delaying the release of Met HCl in the oral gel formulation. The optimized formulation E2-08 consisting of 1.88% Sod ALG, 0.63% HEC, and 1.00% calcium carbonate and two more formulations, E2-10 and E2-12 conformed to USP monograph for extended release. Other physicochemical properties, including floating lag time and duration, viscosity, and pH, measured for each batch and FT-IR spectrometry analysis showed no unexpected interaction between Met HCl and excipients. The current study suggests the potential use of a gastroretentive in situ oral gel for Met HCl helping patient compliance. This study highlights that a systematic approach based on DoE allows the formulation optimization.

## 1. Introduction

Metformin hydrochloride (Met HCl) is the most well-known biguanide, and it is used to treat type 2, non-insulin-dependent diabetes mellitus. While it has been the first choice for treating diabetes for a long time, a high-unit dose (500 mg or 1000 mg) and administration frequency hinder patient compliance. Swallowing a large tablet of Met HCl gives patients discomfort, and a relatively short elimination half-life calls for frequent administration (two to three times a day). Met HCl is mainly absorbed through the upper small intestine with bioavailability of ~50%. The absorption of Met HCl in the large intestine is negligible, and keeping Met HCl in stomach for a prolonged time can improve the bioavailability. The development of gastroretentive Met HCl with extended release provided a higher bioavailability than immediate release formulation. However, a large pill size remains a burden to patient compliance.

In biopharmaceutics classification system (BCS), Met HCl belongs to class III with high water solubility and low intestinal permeability. To enhance the permeability of the drug molecule, we can employ various approaches such as nanoemulsion drug delivery system (DDS), spray freeze drying, fatty acid excipients, self-microemulsifying (SME) DDS. Nanoemulsion DDS such as Pluronics^®^ allows higher solubility and permeability for non-permeable lipophilic drug molecule [1]. It promotes gastrointestinal absorption and decreases inter-subject variabilities [2]. In a rat model, nanoemulsion of castor oil and surfactant tween 80 improved the hypoglycemic effect of Met HCl [3]. Spray freeze drying can be used to improve both solubility and permeability by preparing amorphous solid dispersion. For example, the oral bioavailability of oleanolic acid, a BCS class IV compound with low water solubility and low intestinal permeability, could be improved by preparing spray freeze drying [4]. Mokale and coworkers showed that ethyl cellulose nanoparticles loaded with Met HCl could be prepared by spray freeze drying as a way of preparing sustained release formulation [5]. Long-chain fatty acids, such as capric acid (C10:0), lauric acid (C12:0), and oleic acid (C18:1 (cis Δ9)), increase the transpeithelial transport possibly opening tight junctions [6]. In addition, fatty acid stearate could be used to prepare a controlled-release tablet of Met HCl [7]. SME DDS, comprising oil, surfactant, and cosurfactant, can improve the intestinal permeability by forming microemulsion in the intestine after oral intake [8].

To increase the bioavailability of class III compounds, gastroretentive DDS has been developed to prolong the time of absorption when the short retention time limits the biological availability. Gastroretentive DDS can benefit those drugs that are locally active in the stomach, such as misoprostol [9] and antacid [10]. Those drugs mainly absorbed in the upper gastrointestinal region (L-DOPA [11,12], metformin [13], furosemide [14,15], and riboflavin [16]), the drugs unstable in small and large intestines such as captopril [17], and those drugs with low solubility in high pH (diazepam and chlordiazepoxide [18], verapamil HCl [19]) may benefit from gastroretentive DDS. There exist several systems extending the residence time in the stomach based on bioadhesive, size-increase, floating or sinking in gastric fluids with density control [20]. The bioadhesive system allows the adherence to mucosal surface with various excipients such as polycarbophil, lectin, carbopol, chitosan, carboxymethylcellulose, pectin, gliadin, etc. [21,22,23]. The size-increase system rapidly increases the size of formulation and the passage through the pylorus is hindered [24,25]. The swellable formulation may form matrix system slowly releasing the drug in the stomach [26]. The density of floating system can be prepared by preparing the formulation with density less than that of gastric fluid (1.004~1.010 g/mL), or by adding effervescent mixture [20]. The floating allows the formulation to stay in stomach, sustaining the residence time. In contrast with the floating system, sinking systems use the formulation with high density (2.5~3.0 g/mL), by using the heavy excipients such as barium sulfate, iron, titanium oxide, and zinc oxide, etc. The drawback of the sinking system is the big formulation size [20].

The gastroretentive in situ oral gel system belongs to the floating system, and it has been widely used for antacid drug delivery [27]. The suspension state mixture of ingredients in the in situ oral gel system undergoes gelation in stomach due to the change in pH. Gellan gum and sodium alginate (Sod ALG) are frequently used as polymers, and calcium carbonate, calcium chloride, and sodium citrate are used as cross-linkers. 

Studies have reported a variety of floating DDS for Met HCl mainly of tablet formulation [28]. However, the preparation of oral gel for sustained release of Met HCl has been rarely explored. Considering the large size of Met HCl tablets, gastroretentive oral gel would improve patient compliance. 

In this study, we pursued to prepare a Met HCl in situ oral gel formulation with a short floating lag time and extended release property conforming to USP Dissolution <711>. In situ oral gel was prepared with Sod ALG and calcium carbonate, and hydroxyethyl cellulose was used as a thickener. To derive the optimal composition of the formulation, a systematic approach was attempted based on design of experiments (DoE).

## 2. Materials and Method

### 2.1. Materials

Met HCl was purchased from Farmhispania, S.A. (Barcelona, Spain). Alginic acid sodium salt from brown algae with medium viscosity (Sod ALG), sodium chloride, and hydrochloric acid fuming 37% were obtained from Sigma-Aldrich (St. Louis, MO, USA). Natrosol™ 250 hydroxyethylcellulose (HEC) HHX grade was purchased from Ashland (Covington, KY, USA). Calcium carbonate and sodium carbonate (USP grade) were obtained from Daejung (Siheung, Korea). Gaviscon Double Action was purchased from Oxy Reckitt Benckiser (Seoul, Korea). Purified water was prepared using a Milli-Q^®^ Direct 8 Water Purification System from EMD Millipore (Darmstadt, Germany). HPLC grade solvents were used for high-performance liquid chromatography.

### 2.2. Methods

#### 2.2.1. Preparation of Met HCl In Situ Oral Gel

The pilot study for formulation E1 was first conducted to determine the proper composition of Sod ALG and calcium carbonate: Sod ALG and water indicated in Table 1 were mixed with an overhead mechanical stirrer (400 rpm) on the hotplate set at 70 °C. The dispersed suspension was allowed to cool to 30~40 °C, and calcium carbonate was added as crosslinking agent. Sodium bicarbonate and Met HCl was sequentially added to finish the gel preparation. Based on the pilot study, the composition was varied for Sod ALG, calcium carbonate, and HEC, as indicated in Table 2 (formulation E2). The indicated amount of HEC thickening agent Table 2 was added and dispersed completely right after the mixing of Sod ALG and calcium carbonate.

#### 2.2.2. In Vitro Dissolution Test of Met HCl In Situ Oral Gel

To evaluate the sustained release of Met HCl from each formulation described above, in vitro dissolution test was performed in 900 mL of simulated gastric fluid with pH 1.2 at 37 °C using a USP paddle apparatus. The paddle speed was set at 50 rpm, and 12.5 g of formulation was added to dissolution basket to maintain the shape of gel. Samples, of 5 mL each, were taken at 15, 30, 60, 120, 240, 360, and 480 min [28]. The concentration of Met HCl in each sample was determined by measuring the absorbance at 232 according to USP metformin extended-release dissolution method.

#### 2.2.3. Floating Test of Met HCl In Situ Oral Gel

The in vitro floating test was performed by modifying the previously described method [29]. Briefly, we used 500 mL of simulated gastric fluid, the temperature was set at 37 °C, and the fluid kept being stirred with a magnetic bar (80 rpm). Using a 30 mL syringe without a needle, 12.5 g of prepared gel was injected into the fluid about 15 s to simulate the entrance of gastroretentive formulation into stomach. The floating ability of each gel was assessed by floating lag time and duration. The floating lag time, the time required to emerge on the surface, was recorded in seconds. The floating duration was checked every hour up to 12 h. Gaviscon Double Action was used as a control. The floating test was performed in triplicate.

#### 2.2.4. Physicochemical Characterization of Met HCl In Situ Oral Gel

To evaluate the rheological property of Met HCl in situ oral gel, viscosity of the gel was measured using a DV2TLV viscometer from AMETEK Brookfield (Middleboro, MA, USA), keeping the torque within the range of 10~70% [30]. The measurement was performed in triplicate with 5 min intervals. In addition, the pH of each gel was measured.

To examine the formulation compatibility, Fourier-transform infrared spectroscopy (FTIR) was performed using a JASCO Fourier-Transform Infrared Spectrometer 4100 (Tokyo, Japan) [31].

#### 2.2.5. Design of Experiments

MODDE^®^ 4.0 from Sartorius Stedim (Umeå, Sweden) was employed to identify the design space and conditions for associated factors using DoE. In the pilot study of formulation E1, 3^2^ full factorial design was used to investigate the influence of two variables: percent concentration of Sod ALG (X_1_) and equivalent of calcium carbonate (X_2_) compared to Sod ALG monomer. The percentage drug released in 15, 30, 60, 120, and 240 min were chosen as dependent variables (Y_1_, Y_2_, Y_3_, Y_4_, and Y_5_ each).

For the formulation E2 of Met HCl in situ gel containing HEC, a circumscribed central composite design was used to systematically investigate the effect of each excipients. Three key variables were selected as independent factors: the concentrations of Sod ALG (X_1_), HEC (X_2_), and calcium carbonate (X_3_). The dissolution rate of Met HCl in situ oral gel at 15, 30, 60, 120, and 240 min was defined as response Y_1_, Y_2_, Y_3_, Y_4_, and Y_5_ each. We conducted a partial least squares regression analysis of dissolution rate for each study, using the MODDE^®^ 4.0.

## 3. Results and Discussion

### 3.1. Met HCl In Situ Oral Gel Formulation Containing Sod ALG and Calcium Carbonate: Formulation E1

#### 3.1.1. Dissolution Study of Formulation E1

First, we conducted an in vitro dissolution test of Met HCl in situ oral gel for formulation E1. Design layouts for formulation E1 and corresponding dissolution rates are summarized in Table 3. Partial least squares regression analysis indicated that the more Sod ALG, the longer the time required to release Met HCl. This was statistically significant for 15, 30, 60, and 120 min, but not in the 240 min dissolution rate. The effects plots and analysis of variation (ANOVA) plots are shown in Figure 1. Two-dimensional contour plots (Figure 2) indicate that calcium carbonate, in addition to Sod ALG, delays the dissolution rate slightly. However, the contribution of calcium carbonate in delaying the dissolution of formulation E1 was not statistically significant.

While we could see that Sod ALG delays the dissolution of Met HCl in situ oral gel formulation, none of the batches in formulation E1 conformed to the USP monograph for Met HCl extended-release tablets: 20~40% at 1 h, 35~55% at 2 h, 65~85% at 6 h, and not less than 85% at 10 h.

#### 3.1.2. Floating Study of Formulation E1

As seen in Appendix A, floating test of formulation E1 showed that the oral gel formulations floated within 60 s except E1-02, 03, and 04. The floating lag time for Gaviscon Double Action in the same condition was 26 s. Floating duration was longer than 12 h for all the batches.

#### 3.1.3. Physicochemical Characteristics of Formulation E1

The viscosity of Met HCl in situ oral gel measured in triplicates are summarized in Appendix A, and it ranged between 304.8 and 1283.3 mPa·s. The more Sod ALG is in the formulation, the more viscous was the oral gel. The pH values of formulation E1 ranged between 8.8 and 9.2 (Appendix A), and would not affect the stability of Met HCl [32].

### 3.2. Met HCl In Situ Oral Gel Formulation Containing Sod ALG, Calcium Carbonate, and HEC: Formulation E2

#### 3.2.1. Dissolution Study of Formulation E2

Based on the study of formulation E1, we prepared formulation E2 containing HEC as a thickening agent. Design layouts for formulation E2 and corresponding dissolution rates are summarized in Table 4. Partial least squares regression analysis yielded the statistically significant models for 15, 30, and 60 min (R^2^ > 0.9, Q^2^ > 0.5). For 120 and 240 min, predictive relevance Q^2^ values were low (Q^2^ < 0.5).

As shown in Figure 3A, the effects plots indicate that Sod ALG, HEC, and calcium carbonate negatively affect the dissolution rates: Each component contributes to the extended release of Met HCl. In addition, we could observe the interaction of calcium carbonate and HEC in the plots. ANOVA plots (Figure 3B) for formulation E2 imply that partial least squares regression analysis is valid for 15, 30, 60, and 120 min, whereas statistical significance is lacking for 240 min. Three-dimensional response surface plots are shown in Figure 4.

We could see from the dissolution study that batches E2-08, -10, and -12 conform to the USP monograph, and the better dissolution delay was observed in formulation E2 compared to E1. Figure 5 presents the dissolution rates of each batch in formulation E2.

#### 3.2.2. Floating Study of Formulation E2

As seen in Appendix A, floating test of formulation E2 showed that the oral gel formulations floated within 60 s for batches E2-01 through -05, -09 through -12, and the duration of floating was longer than 12 h for all the batches.

#### 3.2.3. Physicochemical Characteristics of Formulation E2

The viscosity of Met HCl in situ oral gel measured in triplicate are as summarized in Appendix A, and it ranged between 1669 and 7888 mPa·s. Both Sod ALG and HEC enhanced the viscosity of the oral gel. The pH values of formulation E2 ranged between 8.5 and 9.2, similar to formulation E1 (Appendix A).

To see whether Met HCl has any undesirable interactions with Sod ALG and HEC, we analyzed the mixtures of Met HCl using FT-IR spectrometry. As seen in Figure 6, we did not observe any potential interactions or reactions between Met HCl and other excipient components. The amine peak in the 3300~3500 cm^−1^ region and other areas did not show substantial changes.

## 4. Conclusions

In this study, gastroretentive in situ oral gel formulations were prepared for Met HCl. In situ oral gel would be a good alternative to a tablet formulation with extended-release properties as it is easier to swallow. We could see that each excipient contributed to delaying the release of Met HCl. While Met HCl is quite stable in water [33], the stability of the formulation has to be established. Further study including the in vivo experiment would verify the suitability of the in situ oral gel formulation of Met HCl as an alternative.

## Figures and Tables

**Figure 1 pharmaceutics-14-01777-f001:**
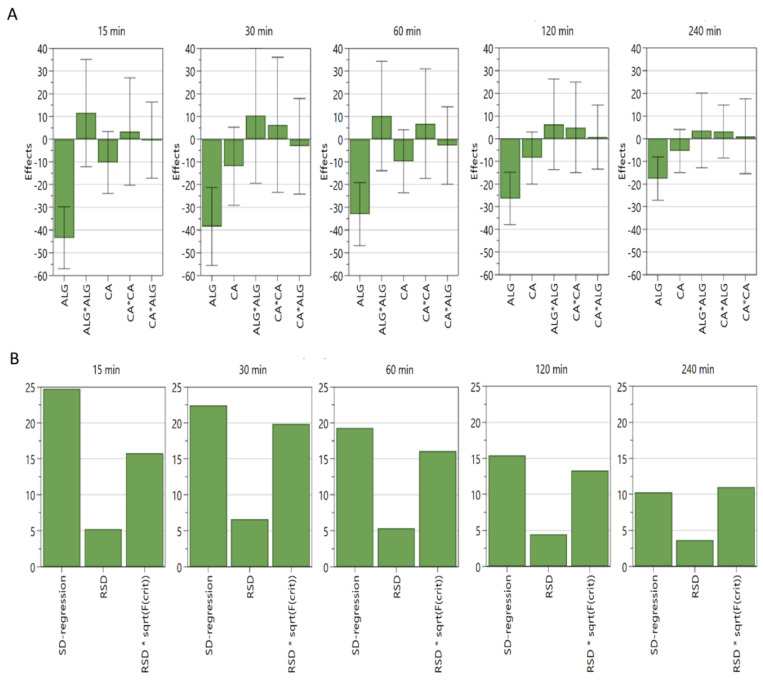
Effects plots and ANOVA plots for formulation E1. (**A**) Effects plots. (**B**) ANOVA plots. N = 9, Degrees of freedom = 3, R^2^ values are 0.97, 0.95, 0.96, 0.95, and 0.93 for 15, 30, 60, 120, and 240 min each in a 95% confidence interval. SD: standard deviation, RSD: relative SD, sqrt(F(crit)): square root of critical F.

**Figure 2 pharmaceutics-14-01777-f002:**
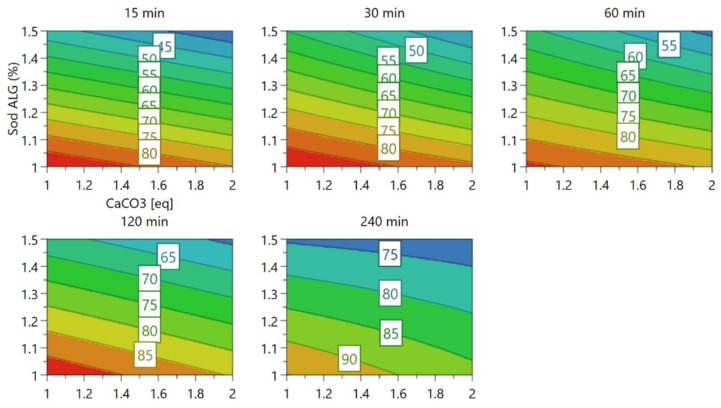
Response contour plots for formulation E1.

**Figure 3 pharmaceutics-14-01777-f003:**
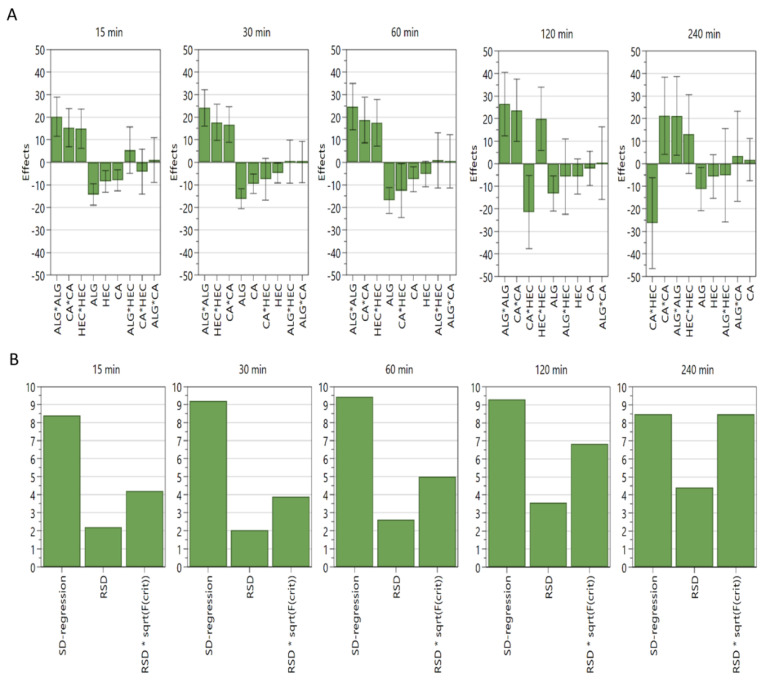
Effects plots and ANOVA plots for formulation E2. (**A**) Effects plots. (**B**) ANOVA plots. N = 17, Degrees of freedom = 7, R^2^ values are 0.95, 0.96, 0.94, 0.90, and 0.83 for 15, 30, 60, 120, and 240 min each in a 95% confidence interval. SD: standard deviation, RSD: relative SD, sqrt(F(crit)): square root of critical F.

**Figure 4 pharmaceutics-14-01777-f004:**
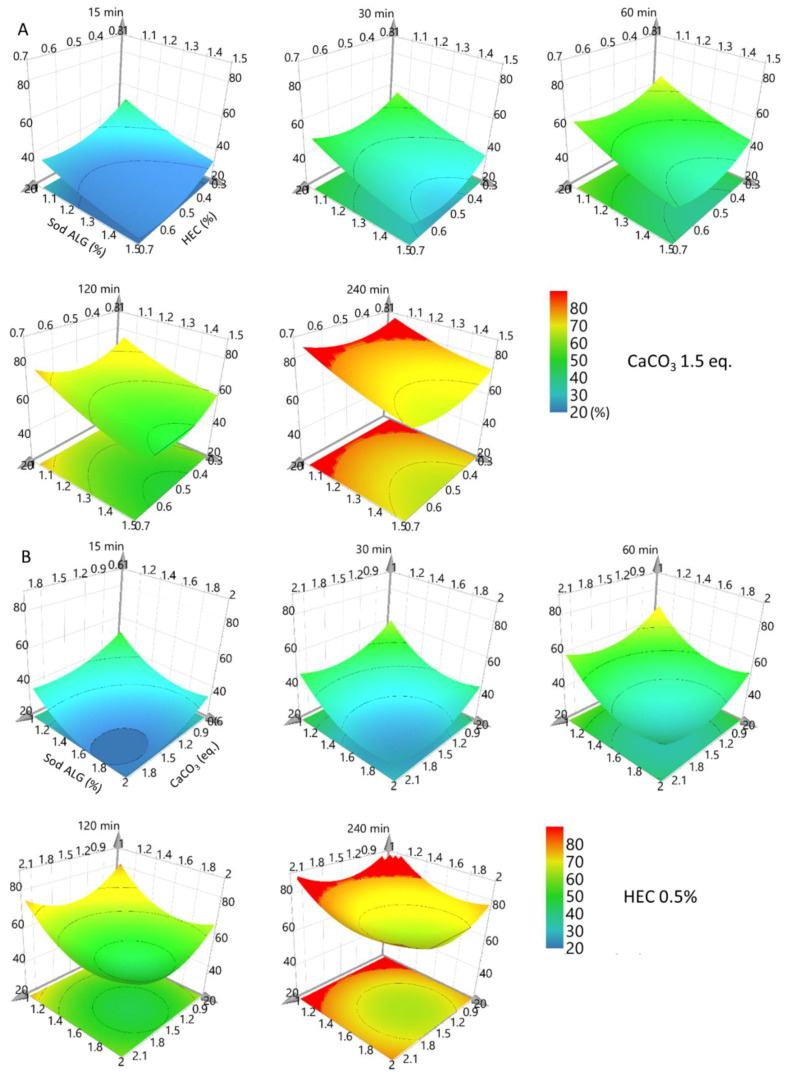
Response surface plots for formulation E2. (**A**) Calcium carbonate = 1.5 eq., (**B**) HEC = 0.5%.

**Figure 5 pharmaceutics-14-01777-f005:**
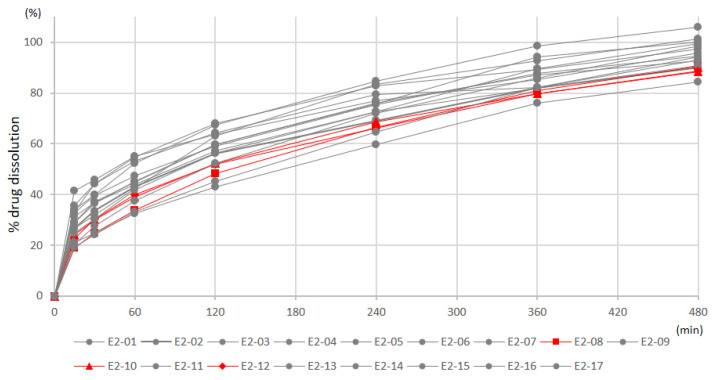
The dissolution profile of each batch in formulation E2.

**Figure 6 pharmaceutics-14-01777-f006:**
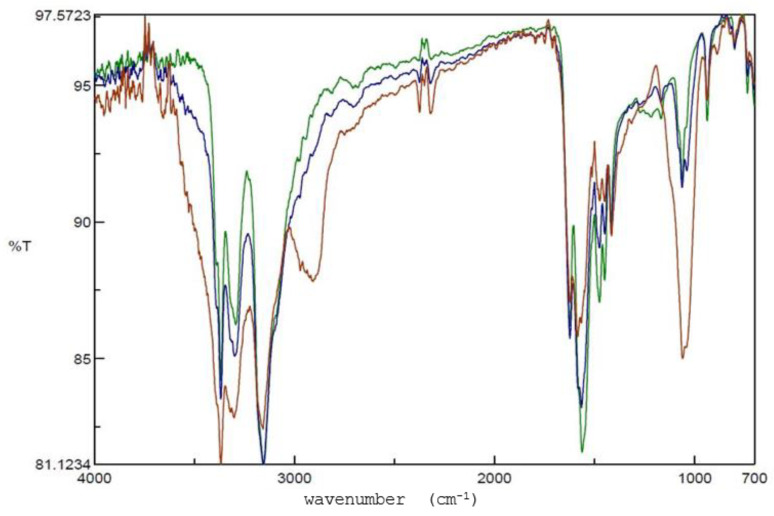
FT-IR spectra of Met HCl. FT-IR spectra of Met HCl alone (green), Met HCl with Sod ALG (blue), and Met HCl with Sod ALG and HEC (brown).

**Table 1 pharmaceutics-14-01777-t001:** Composition of Met HCl in situ gel E1 (in mg).

Batch No.	Met HCl	Sod ALG	Calcium	Sodium	Water	Total
			Carbonate	Bicarbonate		
E1-01	500	125	34	125	11,716	12,500
E1-02	500	125	51	125	11,699	12,500
E1-03	500	125	68	125	11,683	12,500
E1-04	500	156	41	125	11,678	12,500
E1-05	500	156	63	125	11,656	12,500
E1-06	500	156	83	125	11,636	12,500
E1-07	500	188	50	125	11,638	12,500
E1-08	500	188	75	125	11,613	12,500
E1-09	500	188	100	125	11,588	12,500

**Table 2 pharmaceutics-14-01777-t002:** Composition of Met HCl in situ gel containing HEC (E2) (in mg).

Batch No.	Met HCl	Sod ALG	HEC	Calcium	Sodium	Water	Total
				Carbonate	Bicarbonate		
E2-01	500	125	38	33	125	11,679	12,500
E2-02	500	188	38	50	125	11,600	12,500
E2-03	500	125	63	33	125	11,654	12,500
E2-04	500	188	63	50	125	11,575	12,500
E2-05	500	125	38	67	125	11,646	12,500
E2-06	500	188	38	100	125	11,550	12,500
E2-07	500	125	63	67	125	11,621	12,500
E2-08	500	188	63	100	125	11,525	12,500
E2-09	500	104	50	41	125	11,680	12,500
E2-10	500	209	50	84	125	11,533	12,500
E2-11	500	156	29	63	125	11,627	12,500
E2-12	500	156	71	63	125	11,585	12,500
E2-13	500	156	50	26	125	11,643	12,500
E2-14	500	156	50	93	125	11,576	12,500
E2-15	500	156	50	63	125	11,606	12,500
E2-16	500	156	50	63	125	11,606	12,500
E2-17	500	156	50	63	125	11,606	12,500

**Table 3 pharmaceutics-14-01777-t003:** 3^2^ full factorial design layout of formulation E1.

Batch No.	Coded Form		% Drug Released				
	X_1_	X_2_	Y_1_	Y_2_	Y_3_	Y_4_	Y_5_
			15 min	30 min	60 min	120 min	240 min
E1-01	−1	−1	88.5	88.1	90.1	92.3	93.4
E1-02	−1	0	89.9	90.0	91.2	92.6	93.1
E1-03	−1	1	82.6	84.5	87.3	85.9	87.9
E1-04	0	−1	69.9	75.9	78.6	82.7	87.0
E1-05	0	0	56.8	59.9	64.4	70.5	80.4
E1-06	0	1	51.8	53.6	60.8	68.6	75.3
E1-07	1	−1	48.7	56.8	63.2	68.3	74.6
E1-08	1	0	40.0	43.6	51.6	60.1	71.7
E1-09	1	1	41.9	46.9	54.8	63.3	75.4
coded values	actual values						
	Sod ALG (%)	CaCO_3_ (eq.)					
−1	1.00	1.0					
0	1.25	1.5					
1	1.50	2.0					

**Table 4 pharmaceutics-14-01777-t004:** A circumscribed central composite design layout of formulation E2.

Batch No.	Coded Form			% Drug Released				
	X_1_	X_2_	X_3_	Y_1_	Y_2_	Y_3_	Y_4_	Y_5_
	Sod ALG	HEC	CaCO_3_	15 min	30 min	60 min	120 min	240 min
E2-01	−1	−1	−1	41.6	45.8	55.0	63.7	77.0
E2-02	1	−1	−1	27.0	33.6	43.2	56.2	69.1
E2-03	−1	1	−1	33.2	44.2	54.8	68.0	82.7
E2-04	1	1	−1	28.8	36.5	45.2	59.7	75.6
E2-05	−1	−1	1	34.2	39.8	52.4	67.5	84.6
E2-06	1	−1	1	26.6	31.8	42.3	63.2	83.3
E2-07	−1	1	1	29.2	37.1	44.8	59.7	75.9
E2-08	1	1	1	18.9	24.9	33.7	48.2	66.6
E2-09	−1.682	0	0	35.7	44.3	53.1	64.3	79.3
E2-10	1.682	0	0	24.3	30.2	38.9	52.4	68.5
E2-11	0	−1.682	0	31.4	36.8	44.1	57.2	72.5
E2-12	0	1.682	0	22.8	30.4	40.0	52.0	66.0
E2-13	0	0	−1.682	33.0	39.5	47.2	59.1	75.3
E2-14	0	0	1.682	24.7	30.3	41.7	56.3	72.6
E2-15	0	0	0	19.2	24.1	33.2	45.2	64.6
E2-16	0	0	0	20.4	27.6	37.4	52.3	72.1
E2-17	0	0	0	21.0	24.7	32.5	43.0	59.7
coded values	actual values							
	(%)	(%)	(eq.)					
−1.682	1.04	0.29	0.63					
−1	1.25	0.38	1					
0	1.56	0.50	1.5					
1	1.88	0.63	2					
1.682	2.09	0.71	2.25					

## Data Availability

Data are contained within the article.

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
