# Peer review of "Formulation of a Gastroretentive In Situ Oral Gel Containing Metformin HCl Based on DoE"

_pharmaceutics, 2022, doi:10.3390/pharmaceutics14091777_

Round 1

Reviewer 1 Report

The manuscript is well written and the topic is original and interesting. Some experiments are not detailed. It could be accepted after major revision.

Here are some suggestions:

- In the abstract: the cited formulations should be detailed and the results should be better explained. For example, the formulation showing the best pharmacokinetic profile as gastroretentive oral gel should be indicated. 

- In the introduction: the use of DoE procedure and other chemometric methods in the formulation optimization should be described.

- All used methods described in 2.2 need of references.

- In the paragraph 2.2.2, Met concentration has been measured by spectrophotometry at 232. Has the method been developed using a calibration set and validated using a prediction set? Did the authors determine the LOD and LOQ values for this method of determination? Explanations are needed.

-  In the dissolution studies of the formulations E1 and E2, results from partial least squares analysis should be shown. The corresponding graphs should be added in the text or in the supplementary file.

Author Response

We are very grateful for the favorable review of our manuscript, and we took each comment seriously to improve the quality of the manuscript. Please see the attachment in details.

Reviewer 2 Report

The current manuscript deals with in-situ oral gel of metformin hydrochloride for retention in the stomach to aid bioavailability. Overall, the manuscript is well-written. However, there are some queries which need to be addressed before the paper can be published in this journal. Following are the comments.

1.      Page 2 line 61-62 of introduction section requires English correction.

2.      During in vitro floating test, what does 15s mean during injection of gel formulation to the media?

3.      There is no sol-gel study related to the formulation. How did you confirm the conversion of sol into gel at acidic pH? What was the actual sol and gel %fraction?

4.      In Table 3, please remove ‘calcium’ from the actual value at 1 coded value.

5.      In section 3.1.3, line no.180, please remove ‘as’ from the sentence.

6.      On line 220, authors mentioned figure 5 presents dissolution profile, whilst caption in figure 5 mentioned response surface plots. Please correct this. Furthermore, y-axis of figure 5 should be labelled properly, i.e., % drug release or % drug dissolution.

7.      Caption of Figure 6, FTIR spectra of Met HCl (green) should be mentioned instead of only HCl.    

8.      There is no in vivo experiment to back the idea of in situ floating gel. The authors are encouraged to add at least in vivo buoyancy study by adding some contrast agent and present confirmatory images. With this revision, the paper can be published in this journal.

Author Response

(The authors gave the same response as above.)

Round 2

Reviewer 1 Report

In this form, the manuscript can be accepted.

Reviewer 2 Report

The authors have made changes to the manuscript and I accept it now.